# Harmonizing Visual Text Comprehension and Generation

Zhen Zhao[1,2,*], Jingqun Tang[2,✉], Binghong Wu[2], Chunhui Lin[2], Shu Wei[2], Hao Liu[2],
Xin Tan[1], Zhizhong Zhang[1,3], Can Huang[2], Yuan Xie[1,✉]

[1]East China Normal University [2] ByteDance

[3] Shanghai Key Laboratory of Computer Software Evaluating and Testing, Shanghai, China

51255901056@stu.ecnu.edu.cn, tangjingqun@bytedance.com, yxie@cs.ecnu.edu.cn

## Abstract

In this work, we present TextHarmony, a unified and versatile multimodal generative model proficient in comprehending and generating visual text. Simultaneously generating images and texts typically results in performance degradation due to the inherent inconsistency between vision and language modalities. To overcome this challenge, existing approaches resort to modality-specific data for supervised fine-tuning, necessitating distinct model instances. We propose Slide-LoRA, which dynamically aggregates modality-specific and modality-agnostic LoRA experts, partially decoupling the multimodal generation space. Slide-LoRA harmonizes the generation of vision and language within a singular model instance, thereby facilitating a more unified generative process. Additionally, we develop a high-quality image caption dataset, DetailedTextCaps-100K, synthesized with a sophisticated closed-source MLLM to enhance visual text generation capabilities further. Comprehensive experiments across various benchmarks demonstrate the effectiveness of the proposed approach. Empowered by Slide-LoRA, TextHarmony achieves comparable performance to modality-specific fine-tuning results with only a 2% increase in parameters and shows an average improvement of 2.5% in visual text comprehension tasks and 4.0% in visual text generation tasks. Our work delineates the viability of an integrated approach to multimodal generation within the visual text domain, setting a foundation for subsequent inquiries. Code is available at https://github.com/bytedance/TextHarmony.

## 1 Introduction

Visual text comprehension and generation tasks such as scene text detection and recognition [79, 63, 40, 75, 62, 61, 24, 58], document understanding [64, 23], visual question answering (VQA) [26, 15, 27, 39, 59], key information extraction (KIE) [64, 23], multi-modal retrieval[33, 34, 30, 31, 36, 29, 32, 35, 28], visual text generation, editing, and erasure [66, 6, 5] are consistently of significant value for both academic research and practical applications. Recently, remarkable advancements have been achieved in visual text comprehension and generation, driven by the evolution of Multimodal Large Language Models (MLLMs) and diffusion models. Foremost text-centric MLLMs [73, 20, 27, 39] utilize a cohesive framework to comprehend text-rich images comprehensively, whereas diffusion-based approaches [66, 6, 5] introduce innovative modifications to enhance visual text generation capabilities. As depicted in Figure 1, text-centric MLLMs and diffusion models are capable of handling language and vision modalities adeptly, with MLLMs generating texts and diffusion models producing images. However, integrating language and vision generation capabilities within a large multimodal model for visual text scenarios remains unexplored. This paper focuses on

---

* Work done when Zhen Zhao was an intern at ByteDance.  ✉ Corresponding authors.

38th Conference on Neural Information Processing Systems (NeurIPS 2024).

the simultaneous manipulation of language and vision generations to further streamline the processing of diverse text-centric multimodal tasks.

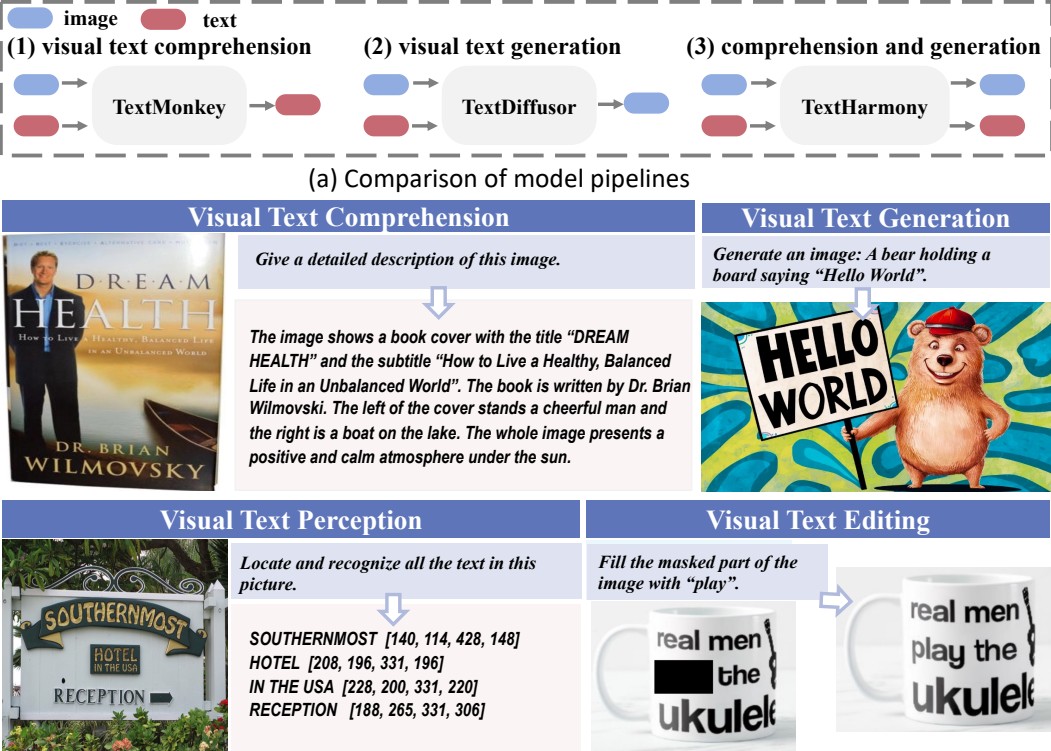

(a) Comparison of model pipelines

(b) Visual text-related multimodal generation tasks

Figure 1: Figure (a) illustrates the different types of image-text generation models: visual text comprehension models can only generate text, visual text generation models can only generate images, and TextHarmony can generate both text and images. Figure (b) illustrates the versatility of TextHarmony in generating different modalities for various text-centric tasks.

In the general multimodal domain, some pioneering efforts [57, 16, 80, 65] empower MLLMs with the ability to generate images beyond texts, vastly extending the versatility of multimodal models. Such advancements inspire us to develop a text-centric multimodal generative model. Our foundational model follows these approaches, incorporating a VIT-based image encoder, a text tokenizer, an LLM, a text detokenizer, and a diffusion-based image decoder.

Previous works [57, 80, 65] and our pilot experiments (Figure 2) have shown that multimodal generation often leads to a notable decline in performance due to the substantial inconsistency between language and vision modalities in the generation space. Prior studies [57, 16, 80, 65] commonly rely on modality-specific supervised fine-tuning to bolster generative capacities. A nuanced challenge involves boosting generative capabilities across modalities using a singular model instance. Mixed-of-Experts (MoE) [54] architecture is widely adopted in LLMs because it improves the model's scalability while keeping the cost of inference similar to that of smaller models. Impressed by MOE-based models [18, 9] that set up task-specific experts to handle different tasks efficiently, we propose adapting modality-specific experts to partially decouple the generation of images and texts. Transforming a dense multimodal generative framework into an MoE-based sparse model poses significant challenges due to the high computational demand and extensive training data requirements.

To tackle these challenges, we incorporate Low-Rank Adaptation(LoRA) [22] experts instead. Specifically, we integrate multiple LoRA experts into the vision encoder and LLM components, encompassing modality-agnostic experts, vision modality generation experts, and language modality generation experts. Modality-specific experts are instrumental in refining and integrating modality-specific generative representations, while modality-agnostic experts enhance certain general representations. A dynamic gating network then amalgamates these modality-specific and general generative rep-

resentations, assigning precise expert weights. Consequently, with minimal parameter increase, we enhance image comprehension and generation capabilities within a singular model instance. Slide-LoRA achieves results comparable to those obtained through separate modality-specific fine-tuning, demonstrating the efficacy of our approach in bridging the gap between language and vision modalities in multimodal generation.

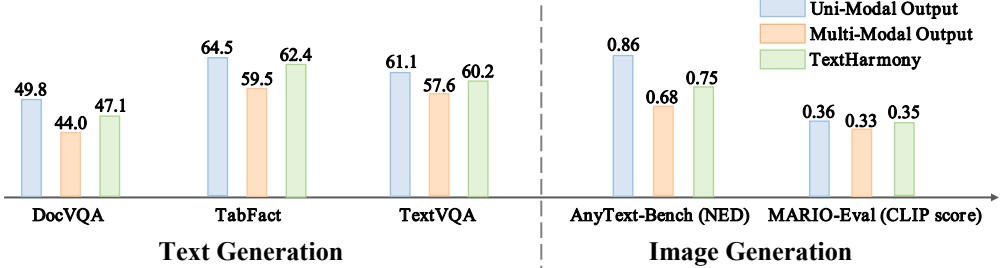

Figure 2: Comparison of single-modal and multi-modal output performance in text generation and image generation Tasks. "Uni-Modal Output" represents the results achieved by modality-specific supervised fine-tuning. "Multi-Modal Output" represents the results achieved by modal-independent supervised fine-tuning. Compared to the multi-modal output, a major performance degradation in the uni-modal output is observed for both text generation and image generation tasks.

Despite the strides made with Slide-LoRA in harmonizing comprehension and generation, the image quality produced by TextHarmony requires improvement. High-quality, detailed image caption data is crucial for visual text-centric image generation tasks. Thus, a detailed image caption dataset, DetailedTextCaps-100K, is created using an advanced closed-source MLLM with prompt engineering. By incorporating DetailedTextCaps-100K in the supervised fine-tuning stage, TextHarmony's image generation quality significantly improves over using original simple captions alone.

Utilizing the above approaches, TextHarmony is versatile in various visual text-centric tasks. In visual text perception tasks, TextHarmony is able to perform well in text detection and recognition, achieving state-of-the-art performance in text grounding tasks. In visual text comprehension tasks, TextHarmony achieves comparable performance to dedicated text comprehension models. In image generation tasks, TextHarmony matches the performance to dedicated visual text generation models. The contribution of this paper can be summarised in three folds:

- We introduce TextHarmony, a versatile large multimodal that allows for the unification of diverse visual text perception, comprehension, and generation tasks. TextHarmony performs comparably to specialized models in visual text perception, comprehension, generation, and editing.

- To mitigate the inconsistency between vision and language modalities in the generative space, we propose Slide-LoRA. Slide-LoRA dynamically aggregates modality-specific and modality-agnostic LoRA experts, partially decoupling the multimodal generative space. With Slide-LoRA, TextHarmony achieves comparable performance to modality-specific fine-tuning results by only adding 2% parameters in a singular model instance.

- A high-quality dataset of detailed visual text image captions (DetailedTextCaps-100K) is constructed with a closed-source MLLM to enhance the performance of visual text generation.

## 2 Related Work

### 2.1 Visual Text Comprehension

Recent multi-modal large language models have increasingly focused on comprehending images with textual information [26, 39, 15, 14, 72, 69, 41, 76, 60, 53, 67]. Among them, UniDoc [15] and mPLUG-DocOwl [72] creates noval text-oriented instruction-following datasets. mPLUG-DocOwl 1.5 [21] builds a parsing system for visual text and employs large-scale pre-training. Monkey [26], DocPedia [14], and HRVDA [27] comprehend dense text by supporting higher image resolution. TextMonkey [39] adopts shifted window attention and filters out significant tokens.

## 2.2 Visual Text Generation

Diffusion-based text-to-image generation has recently achieved impressive progress [19, 50, 77, 52, 48, 7], while the capability of rendering accurate, coherent text in images remains an open problem. DiffUTE [4] designs a text editing model through fine-grained glyph and position information control. GlyphControl [71] generates visual text images by first rendering the glyph and then performing the denoising process. TextDiffuer [6] builds a large-scale text images dataset with OCR annotations and generates visual text conditioned on character-level layouts. Further, TextDiffuer-2 [5] leverages a language model as the layout planner, relieving the character-level guidance. Anytext [66] integrates the text-control diffusion pipeline with an auxiliary latent module and a text embedding module, which has achieved remarkable success in multilingual visual text generation.

## 2.3 Unified Multi-Modal Comprehension and Generation

Current multi-modal large language models (M-LLMs) [82, 2, 1, 11] largely use predicting the next text token as the training objective but exert no supervision for visual data [57]. Recent researches [57, 56, 65, 80, 16, 17, 70] have been attempting to empower M-LLMs to generate visual elements. The Emu family [57, 56] learns with a predict-the-next-element objective in multi-modality and decodes the regressed visual embeddings by a visual decoder. MiniGPT-5 [80] introduces a fixed number of special tokens into the LLM's vocabulary as the generative tokens for images. SEED-LLaMA [16] proposes the SEED tokenizer, which produces discrete visual codes with causal dependency and high-level semantics. MM-Interleaved [65] extracts fine-grained visual details from multiple images' multi-scale feature maps, proving effective in generating interleaved image-text sequences. They all focus on generic multimodal generation, whereas no such work exists yet in the visual text domain.

# 3 Methodology

## 3.1 Model Architecture

Figure 3 presents an overview of TextHarmony. The backbone network follows the paradigm established by MM-Interleaved [65], where a Vision Encoder, an LLM, and an Image Decoder are internally integrated to empower the model to generate both visual and textual content. Specifically, the image embedding extracted by the Vision Encoder is abstracted by a Q-Former [25] and aligns with text tokens. Image and text tokens are then concatenated and forwarded through the LLM. The output token of the LLM either predicts text content or serves as the conditional input of image detokenization. The image decoder perceives the conditional input from LLM and generates images based on the denoising diffusion process [19].

Given the multi-modal input $\boldsymbol{X}$, the multi-modal generation task involves generating interleaved token sequences $\boldsymbol{Y}$, which can be detokenized into both image and text contents. The **token generation stage** is achieved by maximizing the conditional probability under the classic auto-regressive paradigm as follows:

$$P(\boldsymbol{Y}|\boldsymbol{X}) = \prod_{l=1}^{L} p(\boldsymbol{Y}_l|\boldsymbol{X}, \boldsymbol{Y}_{<l}) = \prod_{l \in \mathcal{N}_T} p(\boldsymbol{Y}_l|\boldsymbol{X}, \boldsymbol{Y}_{<l}) \cdot \prod_{l \in \mathcal{N}_I} p(\boldsymbol{Y}_l|\boldsymbol{X}, \boldsymbol{Y}_{<l}), \quad (1)$$

where $\mathcal{N}_T$ and $\mathcal{N}_I$ refer to the index sets of the text and image tokens, respectively. $\boldsymbol{Y}_l$ is the $l$-th predicted token and $\boldsymbol{Y}_{<l}$ is the set of preceding tokens. After that, the text tokens are classified with a linear layer $q(\cdot)$ to produce text content, while the image tokens serve as the condition input in the denoising diffusion process to produce image content. The loss function in the **detokenization stage** consists of the above two parts:

$$\mathcal{L}(\boldsymbol{X}, \hat{\boldsymbol{Y}}) = \underbrace{-(\hat{\boldsymbol{Y}}_{l \in \mathcal{N}_T})^{\mathbf{T}} \cdot log[q(\boldsymbol{Y}_{l \in \mathcal{N}_T})]}_{\text{text generation}} + \underbrace{\mathbb{E}_{\epsilon,t} \|\epsilon - DM(\boldsymbol{Y}_{l \in \mathcal{N}_I}, t)\|^2}_{\text{image generation}}, \quad (2)$$

where $\hat{\boldsymbol{Y}}$ is the ground-truth token sequence and $DM$ is the diffusion model for image denoising.

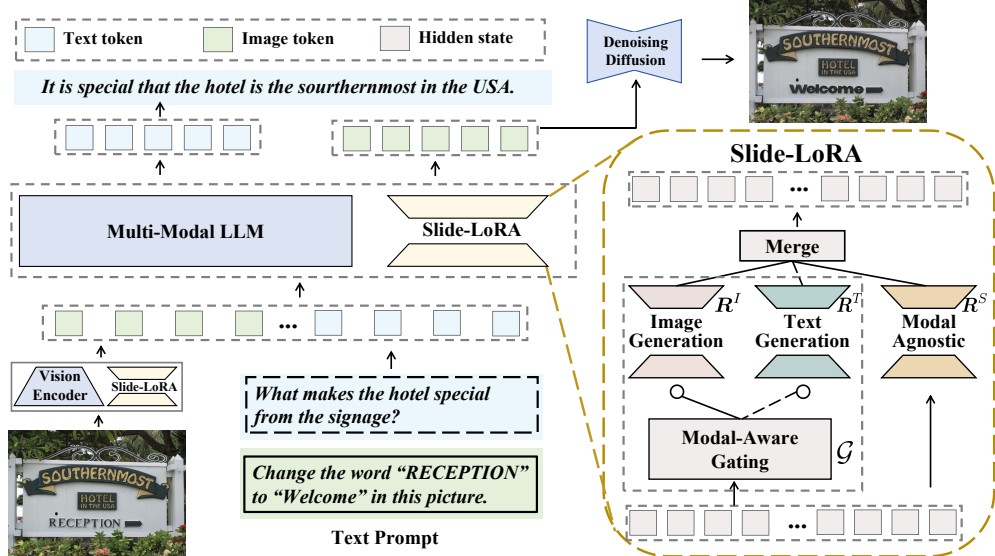

Figure 3: Pipeline of TextHarmony. TextHarmony generates both textual and visual content by concatenating a vision encoder, an LLM, and an image decoder. The proposed Slide-LoRA module mitigates the problem of inconsistency in multi-modal generation by partially separating the parameter space.

On the supervision of Equation 2, the optimization of TextHarmony is tremendously difficult due to inconsistent training objectives. Firstly, text generation aims to generate text from images, while image generation aims to generate images from text, which are mutually exclusive. Secondly, text generation is a classifier problem with a cross-entropy loss function, while image generation (*i.e.*, the denoising process) is a regression problem with a mean square error loss function. To this end, we propose to mitigate the training inconsistency problem by adaptively adjusting the forward pass according to the input tokens.

## 3.2 Slide-LoRA: Enhancing Image and Text Generation Consistency

To harmonize multi-modal generation tasks in a single model, the parameter space would be optimized for inconsistent or conflict training objectives, as stated before. We propose Slide-LoRA (which is shaped like a "slide" between different LoRA experts as shown in Figure 3), a novel module that can be conveniently inserted into Transformer layers as Low-Rank Adaptation (LoRA) [22] and introduces limited parameter increase. As such, Slide-LoRA spontaneously processes text generation and image generation in separate parameter spaces, thus relieving the inconsistent training problem.

As shown in Figure 3, Slide-LoRA is composed of a gating network $\mathcal{G}$ and three LoRA modules $(\boldsymbol{R}^T, \boldsymbol{R}^I, \boldsymbol{R}^S)$. $\boldsymbol{R}^T$ and $\boldsymbol{R}^I$ serve as the separate parameter space for text and image generation, respectively, while $\boldsymbol{R}^S$ aims to learn the knowledge shared by both text and image generation. Specifically, Given the input token sequence $\boldsymbol{x} \in \mathbb{R}^{L \times D}$, the gating network $\mathcal{G}$ (which can be established using either MLPs or rule-based discriminators) determines whether the processing of the input token sequence requires knowledge of text generation or image generation, and produces a scalar $\gamma = \mathcal{G}(\boldsymbol{x}) \in [0, 1]$. The output of the Slide-LoRA layer can be formulated as

$$\boldsymbol{O} = \frac{1}{2} \cdot \{[\gamma \geq 0.5] \cdot \boldsymbol{R}^T(\boldsymbol{x}) + [\gamma < 0.5] \cdot \boldsymbol{R}^I(\boldsymbol{x}) + \boldsymbol{R}^S(\boldsymbol{x})\}, \tag{3}$$

where $[\cdot]$ equals 1 if the condition inside is true and 0 otherwise. Slide-LoRA incorporates task-specific and task-shared knowledge from input tokens, thus separating the inconsistent training objective and learning the shared knowledge of text and image generation.

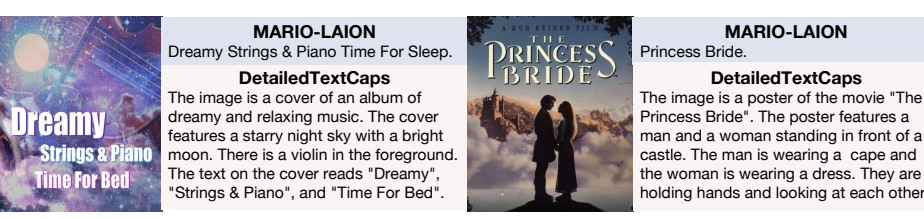

Figure 4: Captions from DetailedTextCaps-100K and MARIO-LAION for the same image. DetailedTextCaps-100K can better depict the textual elements in the image.

## 3.3 Multi-Modal Pre-Training and Comprehensive Fine-Tuning

TextHarmony training process consists of two stages. In the multi-modal pre-training stage, TextHarmony is trained on text-rich image-text corpus and learns to produce multi-modal outputs. In the comprehensive fine-tuning stage, we concurrently cultivate the text and image generation capabilities of TextHarmony by training on a series of text-centric tasks.

### 3.3.1 Stage 1: Multi-Modal Pre-Training

TextHarmony is pre-trained based on the pre-training weight of MM-Interleaved [65], with extra text-rich datasets including MARIO-LAION [6] and DocStruct4M [21]. MARIO-LAION contains 9M web images with brief captions and the according OCR results. DocStruct4M consists of 2M documents and 1M natural images with text-oriented structure annotations. We use MARIO-LAION for both text and image generation (, *i.e.*, either predict the caption of the image or generate the image based on the caption), and we use DocStruct4M for text generation only. In this stage, we freeze the vision encoder and the LLM, training only the Q-Former and the image decoder to obtain basic image understanding and generation capabilities.

### 3.3.2 Stage 2: Comprehensive Fine-Tuning

We integrate various text-centric datasets and employ uniform instructions for all tasks. In this stage, the vision encoder, Q-former, image decoder, and the proposed Slide-LoRA are trained to enhance the multi-modal generation and human-instruction-following capabilities of TextHarmony.

**Visual Text Generation.** In this task, TextHarmony generates images according to the text description and is required to render accurate and coherent text. Although MARIO-LAION contains captions of text-rich images, the description is oversimplified and lacks concentration on the textual elements within the image. To this end, we sample 100K images from MARIO-LAION and generate detailed captions about them, termed DetailedTextCaps-100K. The captions focus on both visual and textual elements in the images. This is achieved by prompting Gemini Pro [12], a pioneer multi-modal large language model, to generate detailed descriptions based on the sampled image and the OCR results. As shown in Figure 4, the image description from DetailedTextCaps-100K is more comprehensive compared with MARIO-LAION and can better depict the textual elements in the image.

**Visual Text Editing.** In this task, TextHarmony substitutes or renders text in the given location of the image and keeps the background consistent. We randomly mask the image with the help of MARIO-LAION's OCR results and fine-tune TextHarmony in a self-supervised manner.

**Visual Text Comprehension.** We employ the training set collected by Monkey [26] for the text-centric VQA fine-tuning. The training set involves 1.4M QA pairs and covers various text-rich scenarios.

**Visual Text Perception.** For the basic OCR capabilities, we randomly sample 1M images from MARIO-LAION and leverage the OCR annotations.

# 4 Experiment

## 4.1 Experimental Setup

**Implementation details.** We use CLIP-ViT-L/14 [49], Vicuna-13B [81], and Stable Diffusion v2.1 [51] as the vision encoder, the LLM and the image decoder, following MM-Interleaved [65]. The image resolution is increased to 896 to capture fine-grained features better. A Q-Former with 12 blocks is adopted to reduce the number of visual tokens to 512. In the multi-modal pre-training stage, the initial learning rate is set to $1e-5$, while in the fine-tuning stage, it is reduced to $5e-6$. The pre-training stage takes 3264 A-100 hours with a batch size of 256; While the fine-tuning stage takes 2352 A-100 hours with a batch size of 64.

**Datasets and Metrics.** We evaluate TextHarmony on a broad range of vision-language tasks. Visual Text Comprehension includes Document-Oriented VQA (InfoVQA [43], DocVQA [44], ChartQA [42]), Table VQA (TabFact [8], WTQ [47]), Scene Text-Centric VQA (TextVQA [55], OCRVQA [45], STVQA [3]) and OCRBench [38]. We adopt the Accuracy metric following TextMonkey [39]. Visual Text Generation includes AnyText-benchmark-EN [66] and MARIOEval [6], where the metric of NED, FID, and CLIP Score are used following AnyText [66] and TextDiffuser [6].

## 4.2 Quantitative Analysis

Table 1: Results of visual text comprehension. TextHarmony is compared with both uni-modal generation models and multi-modal generation models. We employ the Accuracy metric for all methods. TextHarmony* is trained without Slide-LoRA.

| Method | Document-Oriented VQA | | | Table VQA | | Scene Text-Centric VQA | | | OCRBench |
|---|---|---|---|---|---|---|---|---|---|
| | InfoVQA | DocVQA | ChartQA | TabFact | WTQ | TextVQA | OCRVQA | STVQA | |
| *Models for text generation only* | | | | | | | | | |
| LLaVAR [78] | 16.5 | 12.3 | 12.2 | - | - | 41.8 | 24 | 39.2 | 346 |
| UniDoc [15] | 14.7 | 7.7 | 10.9 | - | - | 46.2 | 36.8 | 35.2 | - |
| DocPedia [14] | 15.2 | 47.1 | 46.9 | - | - | 60.2 | 57.2 | 45.5 | - |
| mPLUG-Owl2 [74] | 18.9 | 17.9 | 19.4 | - | - | 53.9 | 58.7 | 49.8 | 366 |
| LLaVA1.5-7B [37] | 14.7 | 8.5 | 9.3 | - | - | 38.7 | 58.1 | 38.1 | 297 |
| Monkey [26] | 25.8 | **50.1** | **54.0** | 49.8 | 25.3 | **64.3** | **64.4** | 54.7 | 514 |
| InternVL [10] | 23.6 | 28.7 | 45.6 | - | - | 59.8 | 30.5 | **62.2** | **517** |
| InternLM-XComposer2 [13] | 28.6 | 39.7 | 51.6 | 62.3 | **28.7** | 62.2 | 49.6 | 59.6 | 511 |
| *Models for both text and image generation* | | | | | | | | | |
| SEED-LLaMA-14B [16] | 23.5 | 6.5 | 13.8 | 49.2 | 13.2 | 14.4 | 16.3 | 20.1 | 357 |
| MiniGPT5 [80] | 2.1 | 1.6 | 1.4 | 4.9 | 0.9 | 2.8 | 2.3 | 2.4 | 68 |
| MM-Interleaved [65] | 17.0 | 8.1 | 11.9 | 40 | 15.1 | 37.2 | 11.7 | 26.4 | 197 |
| TextHarmony* | 26.1 | 44 | 36.2 | 59.5 | 26.1 | 57.6 | 51.9 | 47.2 | 397 |
| TextHarmony-Chat | **28.9** | 49.8 | 38.8 | **64.5** | 28.3 | 61.1 | 57.6 | 51.3 | 448 |
| TextHarmony | 28.5 | 47.1 | 38.0 | 62.4 | 27.1 | 60.2 | 55.3 | 49.7 | 440 |

Table 2: Text grounding performance on MARIO-Eval. The Acc@0.5 metric is employed.

| TGDoc [68] | DocOwl 1.5 [21] | TextHarmony |
|---|---|---|
| 82.5 | 84.3 | **88.7** |

### 4.2.1 Visual Text Comprehension and Perception

**Comparison to Text-Generation and Multi-Modal Generation Methods.** As shown in Table 1, we evaluate TextHarmony on a broad range of visual text comprehension tasks following [39]. As we can see, the performance of TextHarmony is comparable to that of SOTA methods specialized for visual comprehension overall. Specifically, on InfoVQA and TabFact, TextHarmony achieves one of the top performances, with an accuracy of 28.5% and 62.4%. On DocVQA, WTQ and TextVQA, TextHarmony scores 47.1%, 27.1% and 60.2%, which is competitive against top-tier methods like Monkey (50.1%, 25.3% and 64.3%) and InternLM-Xcomposer2 (39.7%, 28.7% and 62.2%). On ChartQA, STVQA, and OCRBench, our model lags behind Monkey, InternVL, and InternLM-XComposer2, which can be attributed to pre-training weights or training data variations. However, TextHarmony surpasses other methods like mPLUG-Owl2 and LLaVA1.5-7B. Furthermore, the performance of state-of-the-art multimodal generative models has a significant performance gap compared to TextHarmony.

**Comparison to TextHarmony\* and TextHarmony-Chat.** To further validate the effectiveness of the proposed method, we train two copies of TextHarmony. TextHarmony\* is trained without the proposed Slide-LoRA module, while TextHarmony-Chat is trained with only Visual Comprehension data and forms the upper bound of TextHarmony on visual text comprehension tasks. As shown in the bottom of Table 1, the performance of TextHarmony\* is strictly lower than that of TextHarmony. For example, TextHarmony scores 3.1% higher on DocVQA, 2.9% higher on TabFact, and 3.4% higher on OCRVQA. Overall, the performance increase brought by Slide-LoRA on comprehension tasks is 2.5%. And the performance gap between TextHarmony and TextHarmony-Chat is thus lowered from 3.96% to 1.5%.

**Comparison on Text Grounding and Recognition.** As shown in Table 2, we evaluate the text grounding capabilities by sampling 300 visual text-position pairs from MARIO-Eval. TextHarmony achieves 88.7% and surpasses current MLLMs with text grounding capabilities (*i.e.,* TGDoc and DocOwl 1.5). The performance on OCRBench (Table 1) reflects the text recognition capabilities of MLLMs. Specifically, TextHarmony surpasses LLaVAR, mPLUG-Owl2, and LLaVA1.5-7B by 94, 74, and 143, while the gap between TextHarmony and top-tier MLLMs remains.

Table 3: Results of visual text editing and generation. TextHarmony is compared with both uni-modal generation models and multi-modal generation models. TextHarmony\* is trained without Slide-LoRA.

|  | NED ($\uparrow$) | FID ($\downarrow$) | CLIP Score ($\uparrow$) |
|---|---|---|---|
| *Models for image generation only* | | | |
| TextDiffuser-2 [6] | 0.81 | 336 | 0.35 |
| GlyphControl [71] | - | 345 | **0.36** |
| AnyText [66] | **0.88** | 352 | **0.36** |
| *Models for both text and image generation* | | | |
| SEED-LLaMA-14B [16] | 0.11 | 348 | 0.27 |
| MiniGPT5 [80] | 0.02 | 380 | 0.25 |
| MM-Interleaved [65] | 0.04 | 412 | 0.29 |
| TextHarmony\* | 0.68 | 356 | 0.33 |
| TextHarmony-Gen | 0.86 | **330** | **0.36** |
| TextHarmony | 0.75 | 342 | 0.35 |

### 4.2.2 Visual Text Generation and Editing

As shown in Table 3, we compare TextHarmony with diffusion model-based text-to-image generation methods and multi-modal generation models. We evaluate the performance of visual text editing by sampling 200 images from AnyText-benchmark-EN and randomly choosing one available text polygon in each image. We use the NED metric to evaluate visual text editing. The performance of Image Generation is evaluated by sampling 100 text-image pairs from MARIO-Eval using the FID and CLIP score metrics. TextHarmony achieves 0.75 on NED and 0.35 on CLIP score, which is comparable to GlyphControl and TextDiffuser. Note that TextHarmony is a multi-modal generation model that is not specialized for image generation and editing, while other multi-modal generation models can hardly generate visual text.

We further train an image generation version of HarmonyText with only image generation/editing data, termed HarmonyText-Gen. As we can see, HarmonyText-Gen achieves 0.86 on NED and 0.36 on CLIP score, surpassing most image-generation methods and achieving comparable results to Anytext. Moreover, the application of Slide-LoRA brings an improvement of 0.07 on NED (from 0.68 to 0.75) and 0.02 on CLIP score (from 0.33 to 0.35), further validating the effectiveness of Slide-LoRA.

### 4.3 Ablation Studies

**Impact of the Config Choice of Slide-LoRA.** As shown in Table 4, we compare the performance of TextHarmony by varying the total number of LoRA modules used in Slide-LoRA. As we can see, the varied number of LoRA modules has little influence on comprehension and generation performance. For example, the accuracy on TextVQA is 60.2%, 60.4%, and 60.4% when $n$ is increased from 3 to 9.

Table 4: Ablation studies of the config choices of Slide-LoRA and the places to insert Slide-LoRA.

| | | Image to Text | | | Text to Image | |
| | | TextVQA | InfoVQA | OCRBench | NED | CLIP Score |
| --- | --- | --- | --- | --- | --- | --- |
| Config Choice of Slide-LoRA | w/o Slide-LoRA | 57.6 | 26.1 | 426 | 0.68 | 0.33 |
| | n=3, s=1 | 60.2 | 28.5 | 440 | 0.75 | 0.35 |
| | n=6, s=2 | 60.4 | 28.2 | 440 | 0.73 | 0.35 |
| | n=9, s=3 | 60.4 | 28.3 | 442 | 0.74 | 0.36 |
| Place to Insert Slide-LoRA | Vision Encoder | 58 | 26.7 | 432 | 0.69 | 0.34 |
| | LLM | 59.9 | 28.1 | 434 | 0.73 | 0.35 |
| | Both | 60.2 | 28.5 | 440 | 0.75 | 0.35 |

Table 5: Ablation studies of DetailedTextCaps.

| DetailedTextCaps-100K | NED | FID ($\downarrow$) | CLIP Score |
| --- | --- | --- | --- |
| w/o | 0.70 | 368 | 0.32 |
| w/ | 0.75 | 342 | 0.35 |

**Impact of varied places to insert Slide-LoRA.** As shown in Table 4, simply inserting Slide-LoRA into the vision encoder would strongly decrease the performance. On the contrary, simply inserting Slide-LoRA into the LLM brings limited performance degradation. The LLM processes multi-modal inputs and generates both visual and textual tokens, while the vision encoder receives only image inputs and generates visual tokens. As a result, when inserted only in the vision encoder, Slide-LoRA would be unable to separate parameter spaces for text generation and image generation.

**Impact of DetailedTextCaps-100K.** We validate the effectiveness of the proposed DetailedTextCaps-100K dataset in Table 5. By removing DetailedTextCaps-100K from the training set and using the original caption from MARIO-LAION, the performance of both the image editing and the image generation decreased. Specifically, the NED decreased from 0.75 to 0.70, and the CLIP score decreased from 0.35 to 0.32. This further validates the effectiveness of detailing the image caption in visual text when training TextHarmony.

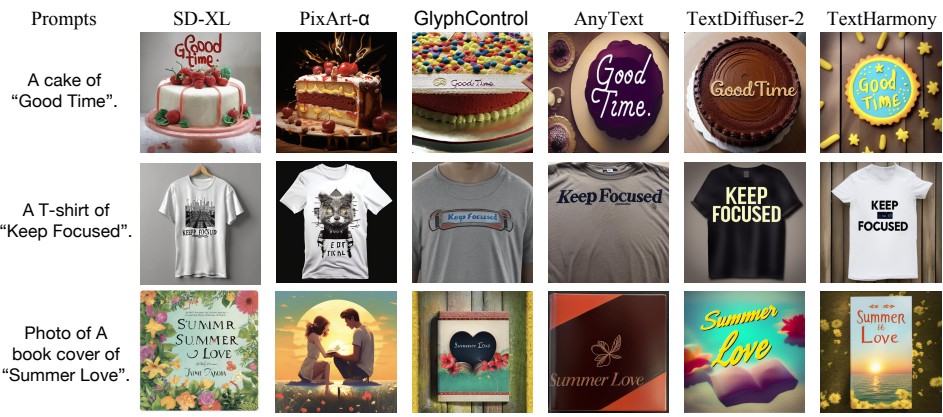

Figure 5: Visualisation of visual text generation.

## 4.4 Qualitative Analysis

We present examples of visual text generation in Figure 5 and examples of visual text editing of TextHarmony in Figure 6.

## 5 Limitation

Despite the strides made by TextHarmony in unifying visual text generation and comprehension within a single model instance, it is important to acknowledge certain limitations of our approach.

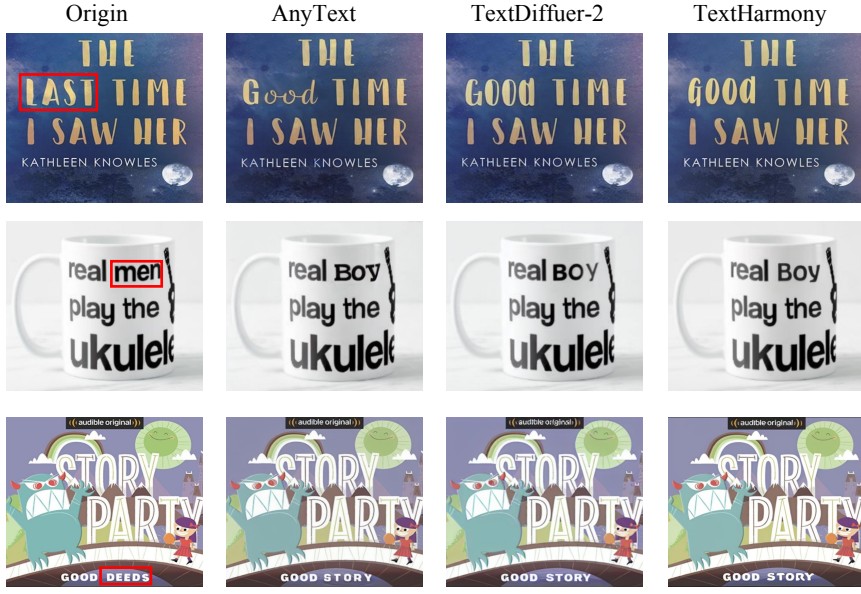

Figure 6: Visualisation of visual text editing.

Firstly, the performance of TextHarmony in visual text perception and comprehension tasks marginally lags behind some of the state-of-the-art open-source models. This discrepancy could be attributed to variations in the training data or the foundational model itself. Secondly, the model's proficiency in generating images is less robust in scenarios characterized by dense text, indicating an area that necessitates further investigation and enhancement. These limitations highlight the need for ongoing research to refine the capabilities of TextHarmony and similar multimodal generative models.

## 6 Conclusion

In summary, this work presents TextHarmony, a versatile multimodal generative model adept at reconciling the disparate tasks of visual text comprehension and generation. Utilizing the proposed Slide-LoRA mechanism, TextHarmony synchronizes the generation process of both vision and language modalities within a singular model instance, effectively addressing the inherent inconsistencies between different modalities. The model architecture is proficient in executing tasks that involve processing and generating images, masks, texts, and layouts, particularly within the realms of Optical Character Recognition (OCR) and document analysis. The accomplishments of TextHarmony herald the significant potential for comprehensive multimodal generative models within the visual text domain. The adaptability of TextHarmony indicates that models of a similar nature could be effectively employed across a diverse array of applications, offering the prospect of revolutionizing sectors that depend on the intricate interplay of visual text comprehension and generation.

**Acknowledgement.** This work is partially supported by National Natural Science Foundation of China (No.62222602, No.62106075, No.62176092, No.62302167, No.62476090, U23A20343), Natural Science Foundation of Shanghai (23ZR1420400), Shanghai Sailing Program (23YF1410500), Natural Science Foundation of Chongqing, China (CSTB2023NSCQ-JQX0007, CSTB2023NSCQ-MSX0137), Chenguang Program of Shanghai Education Development Foundation and Shanghai Municipal Education Commission (23CGA34).

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

# A    Appendix / supplemental material

## A.1    Experiments and Visualisation

### A.1.1    Construction of DetailedTextCaps-100K

DetailedTextCaps-100K is constructed by re-generating the captions about the images from MARIO-LAION through prompting Gemini Pro. Given a sampled image, Gemini Pro is required to generate a detailed caption about the image through the following prompt:

> *Play an image content analysis expert. First, analyze all the image contents in a comprehensive manner and pay special attention to the textual elements in this image. Then,* ***describe the image in as much detail as you can.***

After that, we eliminate the sample if its generated caption is longer than 100 or if the caption contains non-ASCII characters. Further, Advanced MLLMs, including Gemini, are utilized to check the quality of the generated captions through the following prompt:

> *Play an image content analysis expert. First, analyze all the image contents comprehensively and pay special attention to the textual elements in this image. Then,* ***judge whether the caption is consistent with the image and whether the caption can thoroughly describe the image.***

After the above filtering system, we obtained $102,421$ images with high-quality text-oriented captions.

To validate the effectiveness of DetailedTextCaps-100K before merging it into the training set, we let GPT-4V [46] determine which is better, the captions from DetailedTextCaps-100K or the captions from MARIO-LAION. Specifically, we randomly sample 88 image-caption pairs from DetailedTextCaps, and GPT-4V is asked to answer the following question for each image:

> *Which is a better description for this picture? A:<Caption from DetailedTextCaps>, B:<Caption from MARIO-LAION>. Please only answer A or B without any other choices.*

The evaluation result is $82 : 6$, indicating that GPT-4V chooses captions from our DetailedTextCaps-100K as the better image description in most cases. The above evaluation protocol may contain biases from GPT-4V. Thus, we present this result for reference. Nevertheless, the ablation studies conducted in Section 4.3 prove the effectiveness of DetailedTextCaps-100K in favoring visual text generation. More examples of DetailedTextCaps-100K are shown in Figure 7.

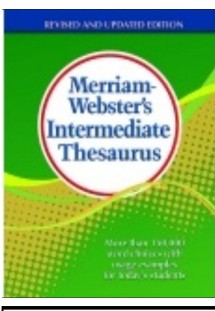
**MARIO-LAION**
Merriam Webster Intermediate Thesaurus Hardcover

**DetailedTextCaps**

The image contains a book titled "Merriam-Webster's Intermediate Thesaurus". The book has a green cover with a yellow and blue circle on the front. The circle contains the words "Merriam-Webster's Intermediate Thesaurus".

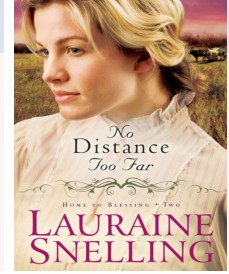
**MARIO-LAION**
No Distance Too Far (Home to Blessing Book #2) - cover

**DetailedTextCaps**
The image shows the cover of a book titled "No Distance Too Far" by Lauraine Snelling. The book is the second in the Home to Blessing series. The cover features a young woman in a white dress with a pink sash standing in a field of wheat. The background is a sunset.

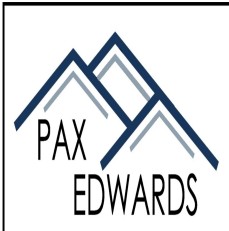
**MARIO-LAION**
Pax Edwards LLC

**DetailedTextCaps**

The image contains a logo of Pax Edwards. The logo is composed of two blue and gray overlapping mountains and the company name "Pax Edwards" below it.The logo is enclosed in a black frame.

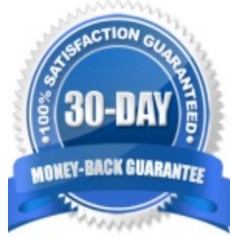
**MARIO-LAION**
30 Day Money Back Guarantee

**DetailedTextCaps**
The image contains a blue and white circular seal. The seal has a thick outer border with the words "100% SATISFACTION GUARANTEED" written in white. Inside the border is a thinner blue ring with the words "30-DAY" and "MONEY-BACK GUARANTEE".

Figure 7: More Examples of DetailedTextCaps-100K.

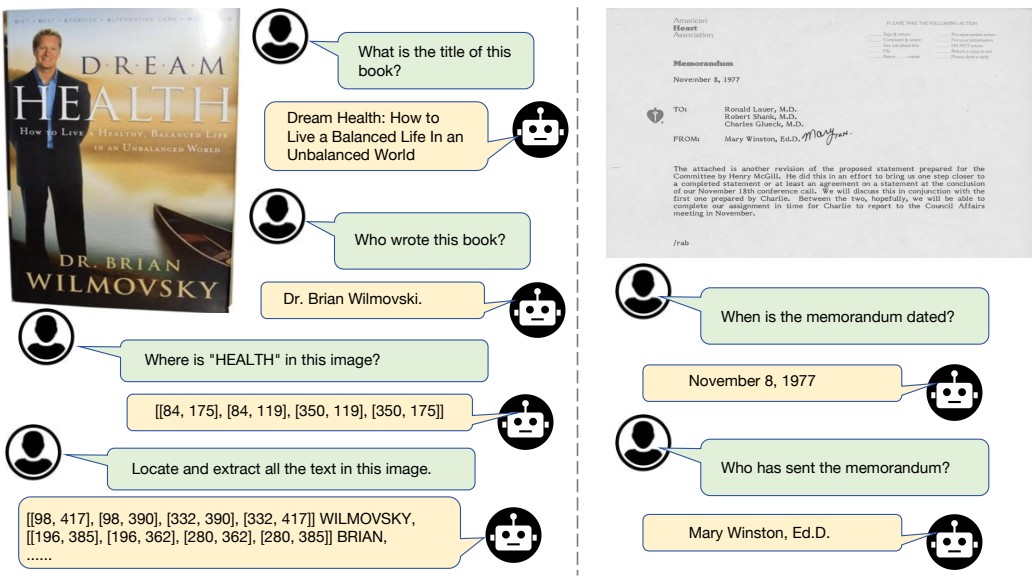

Figure 8: Visualisation of TextHarmony's visual text comprehension and perception capabilities.

### A.1.2 Visualisation of visual text comprehension and perception

Figure 8 presents examples showing the visual text comprehension and perception capabilities of TextHarmony.

Table 6: Prompt design in comprehensive fine-tuning.

| Task | Prompt |
|---|---|
| Visual Text Perception | What is the text in <mask> in this image? |
| | Where is <text> in this image? |
| | Extract all the text in this image. |
| | Locate all the text in this image. |
| | Locate and extract all the text in this image. |
| Visual Text Generation | Generate an image according to the caption. |
| Visual Text Editing | Fill the masked part in this image with <text> |

### A.1.3 Prompt design and data formatting in comprehensive fine-tuning

Table 6 presents the prompt design of TextHarmony in the stage of comprehensive fine-tuning. For visual text comprehension and perception, the input data sequence in the training stage is formulated as:

Answer the following question based on the image. <Image> Question: <Question> Answer: <Answer>.

For visual text generation and editing, the input data sequence in the training stage is formulated as follows:

<Image> <Instruction> <Target Image>.

We employ an all-black image as the input <Image> for visual text generation to make the data formatting consistent with visual text editing. The input data is processed into token sequences by image and text tokenizer, and TextHarmony is trained in the classical auto-regressive manner.

