# OpenReview forum: "Harmonizing Visual Text Comprehension and Generation"
_NeurIPS.cc/2024/Conference — NeurIPS 2024 poster_

### Official Review · Reviewer_LmXP · 2024-07-02

**Soundness:** 3
**Presentation:** 3
**Contribution:** 4
**Rating:** 8
**Confidence:** 5

**Summary:**

This work presents TextHarmony to simultaneously comprehending and generating visual text. The paper performs theoretical and experimental analysis about the performance degradation due to the inherent inconsistency between vision and language modalities. A MoE and LoRA-based module, Slide-LoRA, is then proposed to solve this problem by applying modal-specific and modal-independent LoRA experts dynamically. The experimental results indicate TextHarmony achieves comparable performance to modality-specific fine-tuning results.

**Strengths:**

1. The unification of image understanding and image generation into the visual text domain is a novel approach that broadens the scope of LMM applications.
2. The results presented in this paper looks good and demonstrates the effectiveness of Slide-LoRA.
3. The paper is well organized, with reasonable motivation and insights.

**Weaknesses:**

1. To my knowledge, there exist other multimodal generative models like DreamLLM[1] and Emu[2]. It would be better to compare with these methods.
2. The authors put related work into the supplementary material. It would be better to briefly summarize the background and representitive works in the main text.

[1] Dong, Runpei, et al. "Dreamllm: Synergistic multimodal comprehension and creation." ICLR2024

[2] Sun, Quan, et al. "Emu: Generative pretraining in multimodality." ICLR2024

**Questions:**

1. Could TextHarmony generate more complex glyphs such as Chinese characters?
2. Have you tried using OCR tools to further polish the captions of DetailedCaps-100K? For example, removing the images whose captions are inconsistent with the OCR results.

**Limitations:**

Please refer to weaknesses.

---

> ### Author Rebuttal · Authors · 2024-08-03
>
> Thank you for the valuable comments and the approval of contributions to our work. Your concerns are addressed as follows:
>
>
> **W1: Compare TextHarmony with DreamLLM and Emu.**
>
> In Table E, we compare TextHarmony with DreamLLM and Emu in terms of both visual comprehension and visual generation. As we can see, TextHarmony **performs better** to DreamLLM and Emu in terms of visual text comprehension and generation.
>
>
> > #### Table E: Comparison between TextHarmony, DreamLLM and Emu.
> |    | DocVQA | TabFact | TextVQA | AnyText-Bench | Mario-Eval
> | :---  |   :----:   |  :---: |   :---: |   :---: |   :---: |
> | Emu      | 13.2       | 56.3  | 22.1 |  0.13 |  0.31 |
> | DreamLLM   | 32.7   | 60.4  | 41.8 |  0.18 |  0.30 |
> | TextHarmony   | **47.1** | **62.4**  | **60.2** | **0.75** |  **0.35** |
>
> **W2: Briefly summarize the background and representative works in the main text.**
>
> Thanks for your advice. We will briefly summarize the background and representative works of our paper in the main text of the revised version.
>
> **Q1: Could TextHarmony generate more complex glyphs such as Chinese characters?**
>
> Yes. The training data of TextHarmony in the manuscript only involves English, thus it could not generate Chinese characters. However, the training data of TextHarmony-Align (please refer to Global Author Rebuttal) contains Chinese characters, which makes it capable of generating Chinese characters. We show some examples of Chinese character generation in **Figure C** of the PDF file submitted during rebuttal.
>
> **Q2: Using OCR tools to further polish the captions of DetailedCaps-100K**
>
> Thanks for the suggestion. We use DBNet [1] for text detection and Parseq[2] for text recognition. We filter out the captions that have over 50% mis-matching rate to the OCR results. Specifically, a detected text line is called "mis-matching" if it cannot be found in the caption. After the above procedure, a total of 1839 captions are filtered out from DetailedCaps-100K, suggesting that there is room for improvement in the dataset. Restricted by the rebuttal time, we would like to leave for future work the specific impacts of this strategy to the performance of TextHarmony.
>
> ---
>
> > [1] Real-time Scene Text Detection with Differentiable Binarization. AAAI 2020.
>
> > [2] Scene Text Recognition with Permuted Autoregressive Sequence Models. ECCV 2022.

---

> > ### Comment · Reviewer_LmXP · 2024-08-10
> >
> > I have read the authors' rebuttals and other reviewers' comments, and have a few questions for the authors. The DreamLLM and Emu papers don't include some of the benchmark results in Table E. Did you reproduce them yourselves, and which version of Emu did you use? Besides, in the global rebuttal, TextHarmony-Align outperforms Monkey and Anytext , so was the training data changed from  Mario-Laion to Anytext-3m or was it all used?

---

> > > ### Author Response · Authors · 2024-08-11
> > >
> > > Thanks for your timely reply. Your concerns are addressed as follows:
> > >
> > > **Q3: Did you reproduce them yourselves, and which version of Emu did you use?**
> > >
> > > Yes. We reproduce the results of DreamLLM and Emu based on the official repositories. We use the instruction-tuned Emu, i.e., Emu-I for fair comparison, since TextHarmony has also gone through the instruction tuning stage.
> > >
> > > **Q4: Was the training data changed from Mario-Laion to Anytext-3m or was it all used?**
> > >
> > > The training data was changed from Mario-Laion to Anytext-3m for fair comparison.

---

> > > > ### Comment · Reviewer_LmXP · 2024-08-11
> > > >
> > > > TextHarmony incorporates multimodal generation tasks in visual text, with an effective method called Slide-LoRA proposed towards the modality inconsistency problem. My corncens are well solved. As for the main concerns raised by other reviewers, namely performance comparison to specialized models like AnyText and Monkey, the authors have demonstrated the promising performance of TextHarmony-Align in their Author Rebuttal. Additionally, I agree with their response regarding the comparison baseline. Overall, I believe this work is rather solid and may make a big impact in the field of visual text. I am increasing my rating to 8 consequently.

---

> > > > > ### Author Response · Authors · 2024-08-12
> > > > >
> > > > > Thank you for the appreciation of our work.

---

### Official Review · Reviewer_r95i · 2024-07-02

**Soundness:** 4
**Presentation:** 3
**Contribution:** 3
**Rating:** 8
**Confidence:** 4

**Summary:**

This paper introduces a multimodal generative model (TextHarmony) for unified   comprehension and generation of visual text. To overcome the performance degradation brought by modality inconsistency, the authors propose the slide-lora, which partially decouples the multimodal generation space. An image-text caption dataset, DetailedTextCaps-100K, is also developed to enhance visual text generation capabilities.

**Strengths:**

1. TextHarmony involves visual text coprehension and generation in a single model for the first time. It achieves comparable performance to modal-specific models. It is a solid step forward for multimodal task unification in visual texts.
2. The analysis of the modality inconsistency problem in the multi-modal generation is reasonable and the proposed solution(SlideLoRA) is well motivated, novel and effective.

**Weaknesses:**

a) The connection between this work and visual text is not clearly stated. I understand the focus of this work is to construct a multimodal generative model, but it would be helpful to elaborate specifically how this work achieves multimodal generation in the field of visual text.

b) It would be helpful to report the model size and the inference speed of TextHarmony.

**Questions:**

Line 74 ‘a versatile large multimodal’. Do you mean ‘a versatile large multimodal model’ ?

**Limitations:**

See above

---

> ### Author Rebuttal · Authors · 2024-08-03
>
> Thanks for your time to review our paper,  the valuable comments and the approval of contributions to our work. And we are looking forward to further discussions with you. Your concerns are addressed as follows:
>
>
> **W1: The connection between this work and visual text is not clearly stated. ...**
>
> To improve the performance of visual text comprehension, we increase the resolution of the input images (specifically from 448 to 896). Then, in the pre-training phase, we use images with rich OCR annotations (e.g., DocStruct-4M) to enhancing the text perception abilities of the model. For visual text generation, we randomly mask the text portions of the image and force the model to generate these portions in order to focus training on the generation of text elements.
>
> **W2: It would be helpful to report the model size and the inference speed of TextHarmony.**
>
> Thanks for the suggestion. TextHarmony has **15.4 billion parameters** in total. On AnyText-Bench, the generation of each image costs **1340ms** on average. On DocVQA, the inference time for each output text token is **92ms** on average.
>
> **W3: Line 74 ‘a versatile large multimodal’. Do you mean ‘a versatile large multimodal model’ ?**
>
> Yes. We will modify it in the revised version.

---

> > ### Comment · Reviewer_r95i · 2024-08-12
> >
> > Thanks to the clarification, my concerns are nicely addressed. Like I mentioned within the Strengths, TextHarmony is a solid work with reasonable motivation and impressive experimental results, and I am inclined to accept it.

---

> > > ### Author Response · Authors · 2024-08-13
> > >
> > > We sincerely appreciate your timely feedback and the strong support for our work. We are committed to incorporating all of the clarifications you suggested  in the next version of our paper.

---

### Official Review · Reviewer_2avc · 2024-07-11

**Soundness:** 2
**Presentation:** 3
**Contribution:** 2
**Rating:** 5
**Confidence:** 4

**Summary:**

TextHarmony is a versatile multimodal generative model designed to comprehend and generate visual text. Traditional methods struggle with the inconsistency between vision and language modalities, leading to performance issues. TextHarmony overcomes this with Slide-LoRA, which combines modality-specific and modality-agnostic LoRA experts, ensuring a unified generative process. The model is enhanced by a high-quality image caption dataset, DetailedTextCaps-100K. Experiments show that TextHarmony, with only a 2% increase in parameters, matches modality-specific fine-tuning performance and improves visual text comprehension and generation tasks.

**Strengths:**

+ Explore the possibility of integrating visual text comprehension and generation
+ The proposed Slide-LoRA method is effective to harmonize the training of different modalities and tasks
+ Experimental results show that the combination is possible and the effectiveness of the proposed model

**Weaknesses:**

- The motivation of combing visual text comprehension and generation is not clear. Only the first try (maybe) to integrate the two tasks may not be convincing enough.

- As shown in Table 1 and Table 3, the performance of the proposed method is not superior over existing baselines, e.g., TextHarmony vs. Monkey for comprehension and TextHarmony vs. AnyText for generation. It may be not easy to identify the advantages of combining visual text comprehension and generation. From this point of view, this work seems to simply try combing these two aspects, and not provides valuable research insights.

- Due the limited performance improvement, the human evaluation becomes more necessary to discriminative the proposed method and existing baseline models.

- More explanations or evidence are expected to support certain arguments, e.g., 1) “the optimization of TextHarmony is tremendously difficult due to inconsistent training objectives” in line 106, 2) “mutually exclusive” in line 108, and 3) why classifier and denoising problems are inconsistent as discussed in line 111?

- Some unclear experimental settings, such as “w/o Slide-LoRA”, n and s in Table 4.

- From the qualitative results in Figure 6, it seems that there is not any evident improvement of TextHarmony compared with AnyText.

**Questions:**

Please refer to the above weaknesses.

**Limitations:**

The interleaved generation ability of the proposed approach is unknown.

---

> ### Author Rebuttal · Authors · 2024-08-04
>
> Thanks for your careful review and valuable comments. We are looking forward to further discussions with you. Your concerns are addressed as follows:
>
> **W1: The motivation ... is not clear ... first try ... may not be convincing ...**
>
>
> Our motivation is more than just addressing gaps in unified visual text comprehension and generation.
> A unified model combining visual  text comprehension and generation is indispensable in many aspects:
> - In many scenarios like multi-modal story generation [1], multi-modal document generation [2], the model is required to **generate coherent multimodal outputs**. Separate models cannot guarantee the contextual consistency of multimodal content.
> - Using separate models **increases deployment and maintenance costs**.
> - Recent works like GPT-4o[3] and Chameleon[2] show that the unification of visual and text generation expands the scope of MLLMs and enables a more unified process of multimodal data. Please also refer to reviewer LmXP "S1: The unification ... visual text domain is a novel approach that **broadens the scope of LMM applications**".
>
> Concerning the unsatisfactory performance of previous multi-modal generation models in visual text (refer to Table 1 and 3 in the manuscript), we focus on perception, comprehension and generation of visual text and **address the issue of multimodal generation inconsistency**.
>
>
> **W2 (1): The performance ... is not superior ...**
>
> Please refer to the response to **Common Issue 1** in our Global Author Rebuttal.
>
>
> **W2 (2): ... not easy to identify the advantages of combining ...**
>
> - A unified generation model is indispensable (detailed in response to W1).
> - The modality inconsistence brings performance degradation. We relieves the degradation and "achieves comparable performance to modal-specific models" (refer to S1 of Reviewer r95i).
> - Aligning the settings with Monkey and AnyText, TextHarmony-Align **outperforms Monkey and AnyText** (refer to **Response (2),  Table B and C** in the Global Author Rebuttal).
> - Interleaved multimodal generation abilities(Figure B in the PDF file).
>
> **W2 (3): ... seems to simply try ... not provides valuable research insights.**
>
> Our work is **not** simply trying to combine visual text comprehension and generation.
> - Besides unifying visual text comprehension and generation, we demonstrates that the **inconsistency in the multimodal generative space** is a cause of underperformance.
> - The proposed Slide-LoRA shows 2.5% gains in comprehension and 4.0% in generation tasks, demonstrating its effectiveness against modal inconsistency.
> - Our contribution is also acknowledged by other reviewers ("The analysis of the modality inconsistency problem ... is **reasonable**  ... SlideLoRA is **well motivated, novel and effective**" by Reviewer r95i,  "The results ... **demonstrates the effectiveness** of Slide-LoRA" by Reviewer LmXP).
>
> **W3: Due to limited performance improvement, human evaluation ... necessary ...**
>
> Slide-LoRA achieves 2.5% gains in comprehension and 4.0% in generation tasks compared to baseline. TextHarmony-Align outperforms Monkey and AnyText(**Table B and C** in the **Global Author Rebuttal**). As a result, we think the improvement is **not limited**. Regardless, we agree with you about the importance of human evaluation(**Table D**).
>
> **W4: More explanations or evidence are expected to support certain arguments**
>
> (1): "the optimization ... difficult due to inconsistent training objectives" in line 106
>
> We draw this conclusion mainly from our pilot experiment (Figure 2 in the manuscript). The performance declines a lot (4%~8%) when simultaneously optimizing comprehension and generation, which is also observed in studies like [2] (i.e., the performance of Chameleon-MultiTask is much worse than Chameleon-SFT).
>
> (2): "mutually exclusive" in line 108
>
> Here "mutually exclusive" refers to the fact that text and images are naturally inconsistent in many aspects (information density, data structure, information granularity, etc), thus text generation and image generation require different (exclusive) feature and generation space. Thus, modality alignment has been a fundamental issue in multimodal learning[1, 2, 4].
>
> (3): why classifier and denoising ... inconsistent ... line 111
>
> Previous studies [5,6] show the inconsistency of classification and regression tasks in deep learning. For example, in object detection, a double-head architecture splitting classification and regression has better performance than single-head[5]. In our work, the optimization of TextHarmony contains classification (text generation) and regression (image generation), which are also inconsistent and more difficult given that it is a multi-modal generation issue.
>
> **W5: Some unclear experimental settings, such as "w/o Slide-LoRA", n and s in Table 4.**
>
> "w/o Slide-LoRA" refers to training TextHarmony without Slide-LoRA module. "n" refers to the total number of lora experts in Slide-LoRA. "s" represents RT, RI and RS (Line 120-122), each contains "s" lora experts, i.e., s=n/3. We will add them in the revised version.
>
> **W6: ... there is not any evident improvement of TextHarmony compared with AnyText.**
>
> We would like to clarify that we do not claim the performance of TextHarmony is better than AnyText.  On the issue of comparisons between model performances, please refer to **Response (2),  Table B and C**.
>
> **L1: The interleaved generation ability ... is unknown.**
>
> TextHarmony supports generate interleaved sequences, as showcased in Figure B in the PDF file.
>
> ---
>
> >[1] SEED-Story: Multimodal Long Story Generation with Large Language Model. Arxiv 2024
>
> >[2] Chameleon: Mixed-Modal Early-Fusion Foundation Models.  Arxiv 2024
>
> >[3] Hello GPT-4o. OpenAI, 2024
>
> >[4] Learning transferable visual models from natural language supervision. ICML 2021
>
> >[5] Revisiting the Sibling Head in Object Detector. CVPR 2020
>
> >[6] D2det: Towards high quality object detection and instance segmentation. CVPR 2020

---

> > ### Comment · Reviewer_2avc · 2024-08-13
> > **Thanks for the authors' rebuttal**
> >
> > Thanks for carefully considering my comments and adding the human evaluation results. Some of my concerns have been addressed, e.g., the performance comparison and certain unclear arguments. Therefore, I raise my score to 5.
> >
> > However, compared with the existing work combining comprehension and generation, this work seems not to provide valuable insights (e.g., the synergetic relations in DreamLLM) in terms of research, even though some applications may be promoted by unifying these two.

---

> > > ### Author Response · Authors · 2024-08-13
> > >
> > > We sincerely appreciate your timely feedback and raising your score.  We would like to clarify our research insights.
> > >
> > > Our valuable research insights mainly lie in the word "Harmonizing". In addition to unifying visual text perception, comprehension and generation in a single model, we reveal the issue of modal inconsistency in multimodal generation through comprehensive observation (Emu2 [1], Chameleon [2], MM-Interleaved [3] ) and experiments ( Figure 2 in the manuscript ). The different performance of DreamLLM may be due to the optimization of the model structure and training strategy. We then propose an effective solution, Slide-LoRA, which uses multiple LoRA experts to partially decouple the generative space.  That's why "Harmonizing" rather than "Unifying" is chosen as the title of the work.
> > >
> > > We are open to different perspectives and further discussion. Thank you again.
> > >
> > >
> > > > [1] Generative Multimodal Models are In-Context Learners. CVPR 2024
> > >
> > > > [2] Chameleon: Mixed-Modal Early-Fusion Foundation Models. Arxiv 2024
> > >
> > > > [3] MM-Interleaved: Interleaved Image-Text Generative Modeling via Multi-modal Feature Synchronizer. Arxiv 2024.

---

### Official Review · Reviewer_SovZ · 2024-07-12

**Soundness:** 3
**Presentation:** 3
**Contribution:** 2
**Rating:** 6
**Confidence:** 4

**Summary:**

This work presents TextHarmony, a unified and versatile multimodal generative model proficient in comprehending and generating visual text. Simultaneously generating images and texts typically results in performance degradation due to the inherent inconsistency between vision and language modalities.

**Strengths:**

-  This work introduces TextHarmony, a versatile large multimodal that allows for the unification of diverse visual text perception, comprehension, and generation tasks. TextHarmony performs comparably to specialized models in visual text perception, comprehension, generation, and editing

- The proposed Slide-LoRA dynamically aggregates modality-specific and modality agnostic LoRA experts, partially decoupling the multimodal generative space.

-  A high-quality dataset of detailed visual text image captions (DetailedTextCaps-100K) is constructed with a closed-source MLLM to enhance the performance of visual text generation

**Weaknesses:**

- Details of the Modal-Aware Gating are not given?

-  Results of visual text editing and generation in Table 2 demonstrate that Anytext achieves better performance.

-  The case in Fig. 5, "Good Time" and  "Summer Love", are not correctly synthesized, and the performance is worse than Textdiffuser2 and Anytext.

**Questions:**

see weakness

**Limitations:**

see weakness

---

> ### Author Rebuttal · Authors · 2024-08-03
>
> Thank you for reviewing our paper. If you have any further comments or suggestions, please let us know. Your concerns are addressed as follows:
>
> **W1: Details of the Modal-Aware Gating are not given?**
>
>
> As stated in **Line 123-127**,  the Modal-Aware Gating is an MLP module containing two linear layers. It determines whether the processing of the input token sequence requires knowledge of text generation or image generation. In the multi-modal pretraining stage, the Modal-Aware Gating is trained with $\gamma$=1 in text generation and $\gamma$=0 in image generation, according to the **Equation (3)** in the manuscript. We will update this section in the revised version.
>
> **W2: Results of visual text editing and generation in Table 2 demonstrate that AnyText achieves better performance.**
>
> Please refer to the **Global Author Rebuttal**,  in which we address your concern in detail in three points.
>
> **W3: The case in Fig. 5, "Good Time" and "Summer Love", are not correctly synthesized, and the performance is worse than Textdiffuser2 and AnyText.**
>
>
> Thanks for the careful review. The incorrectly synthesized character cases (i.e., "e" to "E" and "r" to "R") are case-sensitive text generation issues. As observed from Figure 5 of the manuscript, TextDiffuser-2, which uses the same training set to TextHarmony in visual text generation, also fails to generate the correct character cases (i.e., "Keep Focused" to "KEEP FOCUSED"). Given that the same training data (i.e. Mario-Laion) as TextDiffuser-2 is used, we check the training data and find that **some of the training data is case-insensitive**. Besides, aligned with AnyText's training set, TextHarmony is able to **generate correctly case-sensitive text**, as illustrated in Figure A of the PDF file submitted during rebuttal.  For the comparison with AnyText's performance, please also refer to the response in the **Global Author Rebuttal**.

---

> > ### Author Response · Authors · 2024-08-13
> > **Sincere Invitation to Participate in the Discussion**
> >
> > Dear Reviewer SovZ,
> >
> > We would like to extend our appreciation for your time and comments.  Due to the rush in finalizing the writing, some aspects may cause confusion and misunderstanding. Ensuring that the rebuttal aligns with your suggestions is of utmost importance. We are open to further discussions to clarify any remaining questions or concerns. We would greatly appreciate it if you could consider improving the evaluation after reviewing our responses.
> >
> > Thank you very much for your consideration.
> >
> > Sincerely,
> >
> > The Authors

---

> > > ### Comment · Reviewer_SovZ · 2024-08-14
> > > **RE: Sincere Invitation to Participate in the Discussion**
> > >
> > > All my concerns have been addressed, so I raise the rating to WA (6)

---

> > > > ### Author Response · Authors · 2024-08-14
> > > >
> > > > We sincerely appreciate your timely feedback and raising your score. We are committed to incorporating all of the clarifications you suggested in the revised version.

---

### Author Rebuttal · Authors · 2024-08-03

Thanks to the ACs and reviewers for taking time and effort to review our manuscript. And we are also looking forward to further discussion. Below, we would like to address the common issues raised by reviewers.


**Common Issue 1: Performance compared to unimodal generation models, such as TextHarmony *vs.* Monkey in image comprehension and TextHarmony *vs.* AnyText in image generation. (W2 and W3 by Reviewer SovZ,  W2, W3, and W6 by Reviewer 2avc)**

- **Response (1): Unfair comparison between generic models and specialized models**
    - Please kindly note that AnyText and Monkey are unimodal generation specialized models while TextHarmony is a multi-modal generation model. It is **not particularly fair** to simply compare the performance of a specific task **between specialized models and general models**.  For instance, Monkey's performance on scene text recognition, text grounding and text-centric VQA is much worse than specialized models, as shown in Table A.  Besides, in the field of multi-modal generation, the performance of multi-modal generation models (SEED-LLaMA [1], Emu [2], DreamLLM [3], MM-Interleaved [4], Chameleon [5]) is also inferior to unimodal generation specialized models.
    - Thus, when a generic model is compared with a specialized model, the focus should be on overall performance, not just on the performance of a particular task. The overall performance of TextHarmony is **unanimously approved by all the reviewers** (Reviewer SovZ "TextHarmony performs comparably to specialized models", 2avc "Experimental results show...the effectiveness of the proposed model", r95i "achieves comparable performance to modal-specific models”, and LmXP  "The results presented in this paper looks good"). In addition, among multi-modal generation models, TextHarmony achieves much better performance than other models (Table 1 and 3 in the manuscript) in visual text comprehension and generation.

> #### Table A: Comparison of the performance of Monkey and specialized models.
|    | Scene Text Recognition (Union-14M) | Text Grounding (MSRA-TD500)   | Text-centric VQA (DocVQA) |
| :---  |   :----:   |  :---: |   :---: |
| Monkey      | 32.7       | 13.6  | 66.5 |
| Specialized Models   | **85.2** (MAERec [6]) |  **84.9** (DBNet [7]) | **88.4** (ERNIE-Layout [8]) |

---

- **Response (2): Aligning the settings with specialized models and human evaluation**
    - Monkey and AnyText have different settings with TextHarmony in terms of model architecture, training data, etc. To address this case, we conduct ablation experiments **aligning the settings** of training data (AnyWord-3M in AnyText), image resolution (1344*896 in Monkey), LLM (Qwen 7B in Monkey), and model pipeline (two-stage visual text rendering in AnyText). As shown in Table B and C, our model (TextHarmony-Align) slightly **outperforms Monkey and AnyText** in image comprehension and image generation.
   - What‘s more, following the constructive suggestion by Reviewer 2avc (many thanks), we conduct **human evaluation** in visual text editing following the setting established by TextDiffuser. Specifically, the questionnaire consists of 100 cases, which includes two multiple-choice questions: (1) Which of the following images has the best text rendering quality?  (2) Which of the drawn text best harmonizes with the unmasked region? We have collected 10 questionnaires, and the results are shown in Table D. TextHarmony-Align also outperforms AnyText and TextDiffuser2 in human evaluation.

>  #### Table B:  Comparison of the performance on visual text comprehension.
|   | DocVQA| TextVQA | OCRBench |
| :---  |   :----:   |  :---: |   :---: |
| Monkey      | 50.1      | 64.3  | 514 |
| TextHarmony  | 47.1 |  60.2 | 440 |
| TextHarmony-Align | **52.9** |  **64.5** | **523** |


> #### Table C:  Comparison of the performance on visual text generation.
|    | NED| CLIP Score |
| :---  |   :----:   |  :---: |
| AnyText      |  **0.88**   | 0.36  |
| TextHarmony  | 0.75 |  0.35 |
| TextHarmony-Align | **0.88** |  **0.38** |

> #### Table D: Human evaluation of visual text editing
| |TextDiffuser2| Anytext|TextHarmony|TextHarmony-Align
| :---|:---:|:---:|:---:|:---:|
|Q1|699|740|721|**765**|
|Q2|645|691|685|**698**|

---

- **Response (3): Comparison to the actual baseline**
    - Our actual baseline model is the multimodal generative model without the addition of Slide-LoRA (TextHarmony* in Table 1 and 3 in the manuscript). TextHarmony equipped with Slide-LoRA shows an average improvement of **2.5%** in visual text comprehension and **4.0%** in visual text generation tasks. It demonstrates that TextHarmony considerably mitigates the issue of modal inconsistency in multimodal generation, which is what our paper focuses on.

Hopefully, our reply would address the concerns about model performance.

---

>  [1] Making LLaMA SEE and Draw with SEED Tokenizer.  ICLR 2024.

> [2] Emu: Generative Pretraining in Multimodality.  ICLR 2024.

> [3] DreamLLM: Synergistic Multimodal Comprehension and Creation.  ICLR 2024.

> [4] MM-Interleaved: Interleaved Image-Text Generative Modeling via Multi-modal Feature Synchronizer. Arxiv 2024.

> [5] Chameleon: Mixed-Modal Early-Fusion Foundation Models. Arxiv 2024

> [6] Revisiting Scene Text Recognition: A Data Perspective. ICCV 2023.

> [7] Real-time Scene Text Detection with Differentiable Binarization. AAAI 2020.

> [8] ERNIE-Layout: Layout Knowledge Enhanced Pre-training for Visually-rich Document Understanding. EMNLP 2022 Findings.

---

> ### Author Response · Authors · 2024-08-14
> **Thanks to the Reviewers**
>
> Dear Reviewers,
>
> Thanks for your valuable comments and suggestions during the review and discussion period. These feedbacks play a vital role in improving the quality of our work.
>
> As the discussion session is coming to an end, please do not hesitate to reach out to us if you have any additional comments or questions. We are more than willing to clarify your concerns. We greatly value and appreciate having this opportunity to discuss with you.
>
> Yours sincerely,
>
> The Authors

---

### Decision · Program_Chairs · 2024-09-25

**Decision:**

Accept (poster)

**Comment:**

This paper proposed TextHarmony, a new model for visual text understanding and generation, aka, understanding and generating text in images, which is different from prior work for interleaved text and image generation. In addition to the method, the authors also introduce a new dataset particularly for this task. The results seem quite promising. Reviewers had some general concerns about inferior results on unimodal generation compared to other methods, but the AC agrees with the authors that is not the focus of this study, and the reviewers all stated their concerns are well addressed after the rebuttal. As for the connection and discussion with works like DreamLLM, it is a rather minor issue and can be addressed in the camera ready version. The AC thinks this is a solid work for visual text generation and recommends acceptance for this work.